# Is There Any Opportunity to Provide an HBV Vaccine Booster Dose before Anti-Hbs Titer Vanishes?

**DOI:** 10.3390/vaccines8020227

**Published:** 2020-05-16

**Authors:** Rosa Papadopoli, Caterina De Sarro, Carlo Torti, Claudia Pileggi, Maria Pavia

**Affiliations:** 1Department of Health Sciences, University of Catanzaro “Magna Græcia”, Campus Universitario “Salvatore Venuta”, Viale Europa, 88100 Catanzaro, Italy; rosypapadopoli84@gmail.com (R.P.); catedesarro@gmail.com (C.D.S.); pavia@unicz.it (M.P.); 2Department of Medical and Surgical Sciences, University of Catanzaro “Magna Græcia”, Campus Universitario “Salvatore Venuta”, Viale Europa, 88100 Catanzaro, Italy; torti@unicz.it; 3Department of Experimental Medicine, University of Campania “Luigi Vanvitelli”, Via L. Armanni, 5, 80138 Naples, Italy

**Keywords:** baseline anti-HBs levels, booster dose, HBV vaccine, lifelong protection

## Abstract

Whether the primary Hepatitis B vaccination confers lifelong protection is debated. The aim of the study was to assess the effectiveness of booster doses in mounting a protective HBV immune response in subjects vaccinated 18–20 years earlier. The study population consisted of vaccinated students attending medical and healthcare professions schools. A booster dose was offered to subjects with a <10 mIU/mL anti-HBs titer. The post-booster anti-HBs titer was evaluated after four weeks. The subjects with a <10 mIU/mL post-booster anti-HBs titer, received a second and third dose of the vaccine and after one month they were retested. A <10 mIU/mL anti-HBs titer was found in 35.1% of the participants and 92.2% of subjects that were boosted had a ≥10 mIU/mL post-booster anti-HBs titer, whereas 7.8% did not mount an anamnestic response. A low post-booster response (10–100 mIU/mL anti-HBs) was significantly more likely in subjects with a <2.00 mIU/mL pre-booster titer compared to those with a 2.00–9.99 mIU/mL pre-booster titer. The anamnestic response was significantly related to the baseline anti-HBs levels. A booster dose of the HBV vaccine may be insufficient to induce an immunological response in subjects with undetectable anti-HBs titers. A booster dose might be implemented when an anamnestic response is still present.

## 1. Introduction

There is evidence that universal Hepatitis B immunization is an effective and cost-saving intervention that has been responsible for the extraordinary decline of the incidence of infection with Hepatitis B virus (HBV) worldwide [1,2]. Most HBV immunization strategies rely on a primary course of vaccination in infancy, and after a complete vaccination course more than 95% of subjects produce protective anti-HBs titers (>10 mIU/mL) [3]. However, it is debated whether the primary vaccination confers lifelong protection, considering that the anti-HBs titer declines over time in a consistent proportion of vaccinated subjects after 18–20 years from vaccination [4,5,6,7,8,9,10]. It has been found, however, that a substantial proportion of subjects with a low anti-HBs titer express an anamnestic response to an HBV booster, showing that immune memory may persist even if the anti-HBs titer is lower than 10 mIU/mL [11,12]. These considerations have led to suggest that there is no evidence to recommend a booster dose. On the other hand, there are several studies, in particular in an occupational setting where testing of the anti-HBs titer is frequently performed because its decline is not desirable due to the risk of needlestick injuries [13]. These studies suggest the loss of protection in HBV vaccinated subjects that show an undetectable (<2.00 mIU/mL) anti-HBs titer [1,14]. Therefore, the main aim of the study was to assess the effectiveness of booster doses in mounting a protective HBV immune response in relation to the pre-booster anti-HBs titer after 18–20 years from the primary HBV vaccination course.

## 2. Materials and Methods

The study population consisted of the students (undergraduate and postgraduate) attending medical and healthcare professions schools, at the University Magna Graecia of Catanzaro (Southern Italy) and undergoing health surveillance from January 2014 to June 2018. Details regarding the health surveillance program and the inclusion and exclusion criteria for enrollment of individuals have been described in a previous study that investigated the prevalence and persistence of a ≥10 mIU/mL anti-HBs titer after the primary vaccination [8]. Demographic and health status data were collected through the review of medical records completed during the medical surveillance visit. The vaccination status was verified through the immunization records provided by participants. Only subjects who had received a complete course of HBV vaccination ≥13 years before enrollment were included in the survey, whereas students who had received an inadequate interval dose or any additional booster dose of the HBV vaccine, were excluded. Written informed consent was obtained from the participants at the enrollment. According to the Italian Ministry of Health vaccination schedule, subjects born after 1991 received the primary HBV vaccination in infancy, and were defined as “vaccinated in infancy”, whereas subjects born before 1991 received the vaccination during adolescence and were defined as “vaccinated during adolescence”, nevertheless subjects born in 1991 might have been immunized as either infants or adolescents. A booster dose of the monovalent adult HBV vaccine was offered to subjects who were found to have a <10 mIU/mL anti-HBs titer [15]. Students who received a booster dose of the HBV vaccine were asked to return after four weeks to evaluate their anti-HBs titer. Subjects were then classified, according to their anti-HBs titer, into four groups: Group A (<2.00 mIU/mL nondetectable), group B (2.00–9.99 mIU/mL), group C (10–100 mIU/mL), and group D (>100 mIU/mL). Subjects in group A and B were classified to have not shown an anamnestic response, those in group C have shown a low anamnestic response, and those in group D have shown an adequate anamnestic response.

The subjects with a <10 mIU/mL anti-HBs titer after the HBV vaccine booster dose, according to CDC recommendations [15], received a second and third dose of the vaccine after one and six months to complete a second full vaccination course and after one month after the conclusion of the vaccination course, they were retested for the anti-HBs titer. If the anti-HBs titer continued to be <10 mIU/mL after receipt of a complete three-dose series of vaccination the subjects were considered “non-responders”. A flow chart describing the protocol design is reported in Figure 1.

The study protocol was ratified by the Institutional Ethical Committee (“Mater Domini” Hospital of Catanzaro, Italy) (17 March 2016).

### Statistical Analysis

A statistical analysis was developed using the STATA software program, version 14. The outcome of interest was the anti-HBs titer measured after a booster of the HBV vaccine in subjects with a nonprotective anti-HBs titer at 18–20 years after a full course of vaccination. The outcome variable was categorized into three levels: A <10 mIU/mL anti-HBs titer, 10–100 mIU/mL anti-HBs titer, and >100 mIU/mL anti-HBs titer after a booster dose of the HBV vaccine. The univariate analysis was performed using an appropriate test (T-test, χ^2^ test, χ^2^ for trend, Fisher exact test, and ANOVA test) to examine potential associations between the outcome of interest and several explanatory variables: Gender, age at enrollment, age at vaccination, vaccine dose, occupational category, time since vaccination, smoking status, drinking habits, chronic health conditions, and anti-HBs titer at enrollment. A multinomial logistic regression model was also developed and, according to the Hosmer and Lemeshow strategy [16], independent variables for which the *p*-value was 0.25 or less in the univariate analysis, were included in the model.

## 3. Results

During the study period a total of 1398 students were considered for inclusion, with 1374 participants meeting the inclusion criteria. Exclusion criteria pertained to six subjects who received the primary vaccination less than 13 years before enrollment, to two subjects who were born from HBsAg-positive mothers, to one subject who had received an incomplete course of HBV vaccination, to ten that had received an additional booster dose, and to five who had received the vaccine at an inadequate interval dose. The main characteristics of the study population are reported in Table 1. The average age of participants was 24.8 (SD ± 5.07) and the majority were females (67.1%), 53.7% attended undergraduate medical schools, 19.4% reported to suffer of at least one chronic disease, and 7.6% declared taking medications for chronic health conditions. All of them were negative for HBsAg and anti-HBc.

A <10 mIU/mL anti-HBs titer was found in 483 (35.1%) of the study population; of these, 314 (65%) had a pre-booster undetectable (<2.00 mIU/mL) anti-HBs titer and 169 (35%) of 2.00–9.99 (Table 2). A booster dose of the HBV vaccine was inoculated in 320 (66.3%) subjects.

A flow chart reporting the vaccine doses administered to the study population is reported in Figure 2. Among the 231 who returned after four weeks to evaluate the antibody concentration, 213 (92.2%) had a ≥10 mIU/mL anti-HBs titer, and the remaining 18 (7.8%) had a <10 mIU/mL anti-HBs titer. Only 15 out of the 18 subjects with a <10 mIU/mL anti-HBs titer after four weeks, were boosted with a second and third dose to complete a second full vaccination course. Of these, fourteen were tested for the anti-HBs titer and eleven (78.6%) mounted a >100 mIU/mL antibodies concentration one month after the complete cycle of vaccination, and three (21.4%) developed a response between 10 and 100 mIU/mL. 

As reported in Table 3, among the 231 subjects who were tested after the booster dose, the pre-booster anti-HBs titer was undetectable (<2.00 mIU/mL) in 143 (61.9%) and detectable (2.00–9.99 mIU/mL) in the remaining 88 (38.1%); moreover, a significantly higher post-booster titer was found in subjects with a detectable antibody titer at enrollment (Fisher exact = 17.5, 2 df, *p* < 0.001). In particular, sixteen (11.2%) out of the 143 who had a <2.00 mIU/mL pre-booster anti-HBs titer maintained a <10 mIU/mL post-booster anti-HBs titer, whereas only two (2.3%) out of the 88 subjects who had a 2.00–9.99 mIU/mL pre-booster anti-HBs titer maintained a <10 mIU/mL post-booster anti-HBs concentration. Analogously, the post-booster 10–100 mIU/mL anti-HBs titer was more frequent among subjects with a <2.00 mIU/mL pre-booster titer (26.6%) compared to those with a 2.00–9.99 mIU/mL pre-booster titer (10.2%). Finally, a post-booster anti-HBs titer >100 mIU/mL was significantly more frequent in the subjects that had received the primary immunization in infancy (χ^2^ = 16.4, 4 df, *p* = 0.003), whereas a higher proportion of subjects vaccinated during adolescence (17.5%) reported a <10 mIU/mL titer post-booster dose. The seroprotective post-booster titer was significantly more likely in younger subjects at testing (F-test =−2.39, 2 df, *p* = 0.046), in students attending the undergraduate medical schools (χ^2^ = 14.88, 2 df, *p* = 0.001) and in students that had received a pediatric vaccine dose (χ^2^ = 7.84, 2 df, *p* = 0.02).

The multinomial logistic regression model (Table 4), however, highlighted that only having a nondetectable anti-HBs titer (<2.00 mIU/mL) at enrollment, was significantly associated to no anamnestic response (<10 mIU/mL) (RRR = 0.12, 95% CI = 0.03–0.58) and to a low anamnestic response (10–100 mIU/mL) (RRR = 0.26, 95% CI = 0.12–0.59) compared to an adequate anamnestic response after the booster dose (>100 mIU/mL).

## 4. Discussion

According to an up-to-date knowledge of HBV, seroprotection persists for more than 20 years following HBV primary immunization [7,10,17]. However, currently the direct measurement of immune memory is not yet possible and a high percentage of immunized subjects retain immune memory and would have an anti-HBs response upon exposure to HBV [15]. Immunocompetent subjects who achieve ≥10 mIU/mL anti-HBs concentrations after vaccination have nearly complete protection against both acute disease and chronic HBV infection, even if anti-HBs concentrations subsequently decline to <10 mIU/mL [15]. According to current recommendations, while immunocompetent subjects are not entitled to periodic testing to assess anti-HBs levels, testing for anti-HBs after vaccination is recommended to healthcare workers, as well as to students attending medical and healthcare professions schools.

Therefore, this population has been an object of several studies that have been conducted to examine in detail the long-term persistence of immune response to HBV and to booster HBV vaccine. These studies have mostly reached analogous results: (1) After 18–20 years from the primary HBV vaccination a significant proportion of the population (25–50%) has a <10 mIU/mL anti-HBs titer [7,8,9,10]; (2) if the population is disaggregated by age at vaccination it appears that the proportion of subjects with a <10 mIU/mL anti-HBs titer is higher among those vaccinated in infancy compared to those vaccinated in adolescence, regardless of time since vaccination [8]; (3) the persistence of a >10 mlU/mL anti-HBs titer is also related to the dose of HBV vaccine, being more frequent among those who received an adult dose [8,9,10,11,12,13,14,15,16,17,18]; (4) a very high proportion (85–95%) of subjects with a <10 mIU/mL anti-HBs titer that received a booster dose of HBV vaccine mounted a >10 mIU/mL anti-HBs titer one month after the booster [7,9,10]; (5) the post-booster anti-HBs titer is influenced by the pre-booster titer, with higher post-booster titers in subjects with higher pre-booster titers [19]; (6) most of those who maintain a <10 mIU/mL anti-HBs titer after the booster dose and receive a full course HBV vaccine, show a protective HBV immune response [20]. On the basis of these results, most studies have suggested that there is no need for a booster HBV vaccine dose after primary vaccination in infancy, considering that the post-booster response in even more than 90% of subjects suggests persistence of immune memory regardless of anti-HBs titers and that, due to the long incubation time of HBV infection, there is time for an effective immune response to mount up.

All of these results have been confirmed by the present study, but a more detailed observation of the findings and their inclusion in the context of HBV circulation and Hepatitis B incidence in our country may suggest a more complex scenario.

First of all, we found that most of those who have a <10 mIU/mL pre-booster anti-HBs titer have an undetectable titer (<2 mIU/mL) (65%). Moreover, we demonstrated that immune memory, defined by a post-booster humoral response, was found to be significantly higher among sibjects with a detectable pre-booster anti-HBs titer compared to those having an undetectable anti-HBs level (<2 mIU/mL). Indeed, of these, about 11% showed a <10 mIU/mL post-booster anti-HBs titer, compared to only 2.3% of those who showed a 2.00–9.99 mIU/mL pre-booster anti-HBs titer. Therefore, of the overall 7.8% subjects who showed a <10 mIU/mL post-booster anti-HBs titer, the great majority (88.9%) is composed of subjects with a < 2 mIU/mL pre-booster anti-HBs titer. Since all of the subjects that showed a <10 mIU/mL post-booster anti-HBs titer mounted an effective immune response after the secondary full course HBV vaccination, it is plausible to affirm they were not “non-responders” at the first course of HBV vaccination, but more probably they have not shown an anamnestic response since they have lost immunological memory to the HBV vaccine after 18–20 years from vaccination.

The choice to distinguish subjects who showed a >10 mIU/mL post-booster anti-HBs titer into two groups (10–100 mIU/mL and >100 mIU/mL) allowed us to analyze in detail the effect of the booster dose also in subjects who mounted an anamnestic response. Indeed, in one fifth (20.3%) of subjects a 10–100 mIU/mL post-booster anti-HBs titer was detected, and even in this case this result was more frequent among subjects with a <2 mIU/mL pre-booster anti-HBs titer (26.6%) compared to those with a 2.00–9.99 mIU/mL pre-booster anti-HBs titer (10.2%). Indeed, in the group classified as “low responders”, interpretation of a >10 mIU/mL anti-HBs titer after the booster dose could be problematic because it may result from either an anamnestic response or a primary response after the loss of immune memory [21,22].

In an attempt to extrapolate these results to the overall population of HBV vaccinated subjects after 18–20 years from vaccination we may affirm, in a more conservative scenario, that 2.8% of these subjects have lost their immunologic memory and, of these, 2.4% are those with undetectable pre-booster anti-HBs titers and 0.4% are those with 2–9.9 mIU/mL pre-booster anti-HBs titers. If we include also the “low responders” (10–100 mIU/mL post-booster anti-HBs titers), that represent 7.2% of the total population and derive from 5.9% of subjects with undetectable pre-booster anti-HBs titers and from 1.3% of subjects with 2–9.9 mIU/mL pre-booster anti-HBs titers, the overall proportion of unprotected subjects reaches 9.6%, largely composed by subjects with undetectable pre-booster anti-HBs titers (8.3%).

Within this context, it should be also noted that, although the acute HBV incidence has significantly declined in Italy, 212 cases have been reported in 2018, with an incidence of 0.8 per 100,000, and 16 cases have been observed in vaccinated subjects, five of whom have apparently received a correct full course schedule [23].

It has also been reported that “the meaning of the lack of immune memory, as determined by the failure to develop an anamnestic response following an HBV challenge dose, remains an open question.” [24]. In other words, it cannot be excluded that subjects that do not mount an anamnestic response after the booster HBV vaccine dose are still protected from the HBV clinical disease.

However, on the basis of our results, we believe that it seems crucial to reduce, as much as possible, the proportion of vaccinated subjects who show undetectable anti-HBs titers, since these subjects are at higher risk of having lost immunologic memory, and represent in our context about 22% of the vaccinated subjects after 18–20 years from vaccination. Since it seems that the longer the persistence of a low or undetectable anti-HBs titer the higher the risk of immunologic memory loss [25], a precautionary measure could be to introduce a booster dose before the anti-HBs titer vanishes and becomes undetectable. Cost-effectiveness studies analyzing the opportunity of the offering of a booster dose of the HBV vaccine are warranted.

The principal limitations of our study were as follows. First, the lack of post-primary series anti-HBs titers results does not allow a direct comparison with those obtained after the booster HBV vaccine dose. Moreover, since to distinguish an anamnestic response from a primary response after the loss of immune memory it is necessary to check the anti-HBs titer 7–10 days after the booster HBV vaccine dose [22], another limitation of our study is that, in line with the Italian Legislative Decree 81/2008 [26] that sets some general obligations for workers’ health monitoring, we evaluated the anti-HBs titer after four weeks and, therefore, we were not able to reveal the early anti-HBs seroconversion that could be defined as the presence of immune memory [22]. Third, our study was not designed to assess the cell mediated immune response to HBV vaccination and studies focused on this topic would be complementary and important, especially in those with an undetectable anti-HBs titer that fail to respond to a booster dose.

## 5. Conclusions

In conclusion, the results of this study demonstrated that the response to the booster dose was significantly related to the baseline anti-HBs levels. A booster dose of the HBV vaccine may be insufficient to induce an immunological response in a consistent proportion of subjects who had received a primary HBV vaccination but had undetectable anti-HBs titers. As immunological memory may wane, the booster dose, if cost-effectiveness studies will highlight the opportunity to offer one, should be done when an anamnestic response is still present, which would most easily be achieved during school attendance [27].

## Figures and Tables

**Figure 1 vaccines-08-00227-f001:**
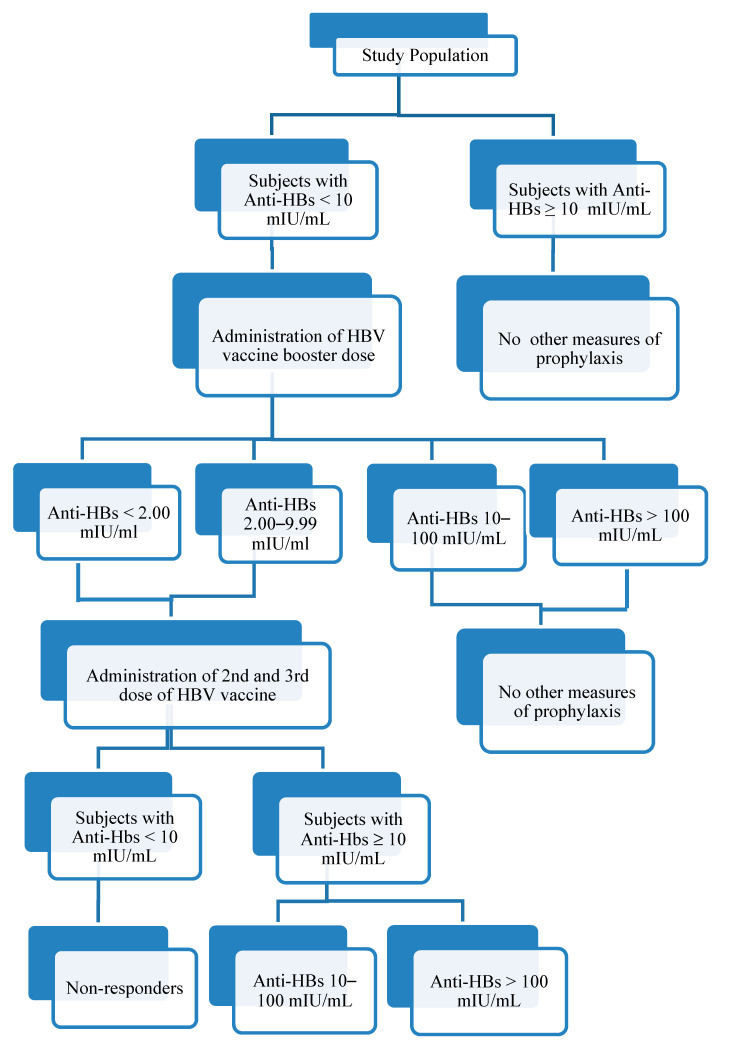
Study protocol flow chart.

**Figure 2 vaccines-08-00227-f002:**
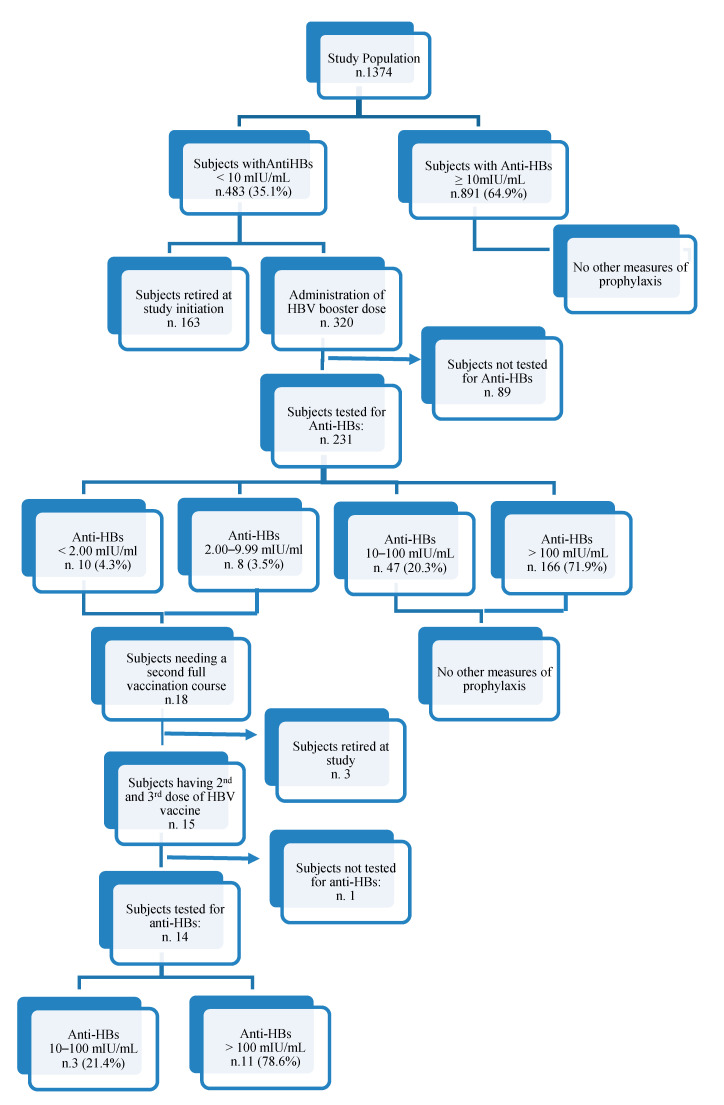
Study results flow chart.

**Table 1 vaccines-08-00227-t001:** Demographic and HBV vaccination characteristics of the study population.

Characteristic	Total	
	N (1374)	%
**Gender**		
Male	452	32.9
Female	922	67.1
**Age at testing, years**		
Mean ± SD	24.8 ± 5.07	
**Attended course**		
Undergraduate	738	53.7
Postgraduate	636	46.3
**Chronic health condition**		
Yes	266	19.4
No	1108	80.6
**Medication for chronic health condition**		
Yes	105	7.6
No	1269	92.4
**Age of HBV vaccination**		
In infancy (0–3 years)	760	55.3
During adolescence (11–14 years)	570	41.5
Other ages	44	3.2
**Time since vaccination, year**	Mean ± SD 19.3 ± 2.8	

SD: Standard deviation. In brackets the number of eligible subjects.

**Table 2 vaccines-08-00227-t002:** Booster dose response stratified by the pre-booster anti-Hbs titer.

Anti-HBs Titer at Enrollment (mIU/mL)	Subjects with <10 mIU/mL at Enrollment	Post-Booster Anti-HBs Titer (mIU/mL)			
	n (%)	<10 n. (%)	10–100 n. (%)	>100 n. (%)	Total n. (%)
**Total subjects**					
<2.00	314 (65)	16 (11.2)	38 (26.6)	89 (62.2)	143 (61.9)
2.00–9–99	169 (35)	2 (2.3)	9 (10.2)	77 (87.5)	88 (38.1)
Total	483 (100)	18 (7.8)	47 (20.3)	166 (71.9)	231 (100)
**Vaccinated in infancy**					
<2.00	243 (64.3)	6 (5.5)	27 (24.8)	76 (69.7)	109 (60.2)
2.00–9–99	135 (35.7)	2 (2.8)	7 (9.7)	63 (87.5)	72 (39.8)
Total	378 (100)	8 (4.4)	34 (18.8)	139 (76.8)	181 (100)
**Vaccinated during adolescence**					
<2.00	60 (65.9)	7 (28)	9 (36)	9 (36)	25 (62.5)
2.00–9–99	31 (34.1)	-	2 (13.3)	13 (86.7)	15 (37.5)
Total	91 (100)	7 (17.5)	11 (27.5)	22 (55)	40 (100)
**Vaccinated in other ages**					
<2.00	11 (78.6)	3 (33.3)	2 (22.2)	4 (44.5)	9 (90)
2.00–9–99	3 (21.4)	-	-	1 (100)	1 (10)
Total	14 (100)	3 (30)	2 (20)	5 (50)	10 (100)

**Table 3 vaccines-08-00227-t003:** Characteristics of subjects tested after the booster dose stratified by a post-booster anti-HBs titer.

Characteristics	Total		Post-BoosterAnti-HBs					
			(<10 mIU/mL)		(10–100 mIU/mL)		(>100 mIU/mL)	
	N (231)	%	N (18)	7.8%	N (47)	20.3%	N (166)	71.9%
**Anti-HBs at enrollment**								
<2.00	143	61.9	16	11.2	38	26.6	89	62.2
2.00–9.99	88	38.1	2	2.3	9	10.2	77	87.5
**Fisher exact test = 17.5, 2 df, *p* < 0.001**								
**Gender**								
Male	81	35.1	5	6.2	23	28.1	53	65.4
Female	150	64.9	12	8.1	24	16.1	113	75.8
**χ^2^ = 4.89, 2 df, *p* = 0.087**								
**Age at testing, years**								
Mean ± SD	22.8 ± 4.1		25.4 ± 4.2		23.6 ± 4.9		22.3 ± 3.7	
**F = −2.39, 2 df, *p* = 0.046**								
**Attended course**								
Undergraduate students	178	77.1	8	4.5	33	18.5	137	77
Postgraduate students	53	22.9	10	18.9	14	27.0	29	55.7
**χ^2^ = 14.88, 2 df, *p* = 0.001**								
**Smoking status**								
Never smoker	183	79.2	16	8.7	38	20.8	129	70.5
Former smoker	4	1.7	-	-	1	25.0	3	75.0
Current smoker	44	19.1	2	4.5	8	18.2	4	77.3
**Fisher exact test =1.51, 4 df, *p* = 0.832**								
**Drinking habits**								
Do not drink alcohol	86	37.2	10	11.6	12	14.0	64	74.4
Rarely/occasionally	145	62.8	8	5.5	35	24.2	102	70.3
Often/daily	-	-	-	-	-	-	-	-
**χ^2^ = 5.46, 2 df, *p* = 0.065**								
**Chronic health condition**								
Yes	55	23.8	5	9.1	8	14.5	42	76.4
No	176	76.2	13	7.4	35	22.2	124	70.4
**χ^2^ = 1.55, 2 df, *p* = 0.464**								
**Age of HBV vaccination**								
In infancy (0–3 years)	181	78.4	8	4.4	34	18.8	139	76.8
During adolescence (11–14 years)	40	17.3	7	17.5	11	27.5	22	55.0
Other ages	10	4.3	3	30.0	2	20.0	5	50.0
**χ^2^ = 16.40, 4 df, *p* = 0.003**								
**Vaccine dose**								
Pediatric	198	85.7	12	6.1	38	19.2	148	74.7
Adult	33	14.3	6	18.2	9	27.3	18	54.5
**χ^2^ = 7.84, 2 df, *p* = 0.020**								
**Time since vaccination**								
≤19 Years	117	50.6	9	7.7	27	23.1	81	69.2
≥20 Years	114	49.4	9	7.9	20	17.5	85	74.6
**χ^2^ = 1.10, 2 df, *p* = 0.577**								

SD: Standard deviation; df: Degree of freedom.

**Table 4 vaccines-08-00227-t004:** Results of the multinomial regression analysis estimating predictors of the anti-HBs titer measured after a booster of HBV vaccine.

Log Likelihood = −153.55; χ^2^ = 44.15 (8 df); *p* = 0.0001; No. of Observation = 231				
Outcome: Anti-HBs Titer Measured after a Booster of HBV Vaccine	Post-Booster anti-HBs <10 mUI/mL		Post-Booster Anti-HBs 10–100 mUI/mL	
	RRR (95% CI)	*p*-Value	RRR (95% CI)	*p*-Value
Anti-HBs titer at enrollment (<2.00 mIU/mL as reference)	0.12 (0.03–0.58)	0.008	0.26 (0.120.59)	0.001
Gender (male as reference)	1.32 (0.38–4.57)	0.664	0.54 (0.261.09)	0.087
Age at enrollment, continuous	1.09 (0.89–1.35)	0.399	1.03 (0.891.18)	0.695
Drinking habits (do not drink alcohol as reference category)	0.39 (0.13–1.15)	0.089	1.42 (0.663.06)	0.367

RRR: Relative risk ratio.

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
