# Peer review of "Is There Any Opportunity to Provide an HBV Vaccine Booster Dose before Anti-Hbs Titer Vanishes?"

_vaccines, 2020, doi:10.3390/vaccines8020227_

Round 1
Reviewer 1 Report
AUTHORS
Manuscript ID: vaccines-802474
Title: Is there any opportunity to provide an HBV vaccine booster dose before anti-HBs titer vanishes?
This is a very short but meaningful study assessing the effectiveness of booster doses in mounting a protective HBV immune response in subjects vaccinated 18-20 years earlier (students). Interesting little study that is very straightforward and presents valuable data. I have comments below
Authors state (introduction) what is considered protective anti-HBs titers (>10mIU/mL). Please reference this
The same here”Immune memory may persist even if anti-HBs titer is lower than 10mIU/mL”. please reference
Please check flowchart that refers 2º and 3º dose where it should be 2nd and 3rd the same on flowchart 2 (results)
On table 1 please define SD. The same with df on table 2 and RRR on table 3
There is an arrow on the results flowchart, at the bottom
Author Response
Reviewer 1:
Author state (introduction) what is considered protective anti-HBs titers (>10 mIU/mL). Please reference this.
As requested we have added the reference in the Introduction section (page 1, line 43) and in the References section (page 3, lines 298-299).
The same here “Immune memory may persist even if anti-HBs titer is lower than 10 mIU/mL”. please reference.
As requested we have added the references in the Introduction section. (page 2 , lines 45 and 48) and in the References section (page 3, lines 300-306).
Please check flowchart that refers 2° and 3° dose where it should be 2nd and 3rd the same on flowchart (results)
As suggested, in Figures 1 and 2 we have replaced “2° and 3°” with “2nd and 3rd”
On table 1 please define SD. The same with df on table 2 and RRR on table 3.
As requested, in table 1 we have defined SD (page 5, line 124). Table 2 was modified accordingly with the comment of the other reviewer and now it was renumbered as table 3 in which we have defined “SD” and “df” (page 9, line 161); in table 3 (now table 4) we have defined RRR (page 10, line 180).
There is an arrow on the results flowchart, at the bottom
In response to the point, we have eliminated the arrow on the results flowchart, at the bottom.
Reviewer 2 Report
Papadopoli et al. have carried out an interesting study in which a significant number of vaccinated subjects have undetectable anti-Hbs titers. The need to provide booster dose in subjects with an anamnestic response is therefore considered. The study is well written, has clear objectives and is methodologically well designed.
Minor changes:
1) In Table 1, “age of HBV vaccination” the sum of subjects is not 1374.
2) A different table would be recommended for subjects with low titers (<10 mIU / mL; n = 483). This additional table would show: “pre-booster anti-Hbs titer”, “age of the subjects depending on the titers”, “Boosted subjects”, etc…
3) It is obvious that the study has clearly discriminated between subjects who have never been vaccinated and those with undetectable titers. In this sense, no subject was excluded due to lack of immunization records? Were the immunization records self-reported by the subjects or can they be accessed through medical records?
Author Response
Reviewer 2:
- In Table 1, “age of HBV vaccination” the sum of subjects is not 1374.
As suggested we have reported the correct number in the Table 1 (page 6).
- A different table would be recommended for subjects with low titers (< 10 mIU/mL; n=483). This additional table would show: “pre-booster anti-HBs titer”, “age if the subjects depending on the titers”, “Boosted subjects”, etc…
As suggested we have added an additional table reporting data about booster dose response in subjects with anti-HBs titer <10 mIU/mL, stratified by pre-booster anti-Hbs titer (Table 2, page 7).
- It is obvious that the study has clearly discriminated between subjects who have never vaccinated and those with undetectable titers. In this sense, no subjects was excluded due to lack of immunization records? Were the immunization records self-reported by the subjects or can they be accessed through medical records?
In Italy, medical surveillance is provided, at no cost, through occupational health physicians for all workers (students included) who are at risk of occupational exposure to blood-borne pathogens, chemicals, or physical agents (noise, radiations). Each workers attending medical surveillance programme during the medical surveillance visit must provide an official Vaccination schedule showing immunizations status. Vaccination schedule get included as a permanent part of the workers' clinical record. Therefore none subject was excluded due to the lack of immunization record. Also, in the Methods section, we have specified that vaccination status was verified through the immunization records provided by participants during the medical surveillance visit (page 2, lines 64-65).